# Masked Structural Growth for 2x Faster Language Model Pre-training

Yiqun Yao[1], Zheng Zhang[1], Jing Li[2], and Yequan Wang[*1]

[1]Beijing Academy of Artificial Intelligence, Beijing, China
[2]Harbin Institute of Technology, Shenzhen, China
{yqyao,zhangzheng}@baai.ac.cn,
jingli.phd@hotmail.com, tshwangyequan@gmail.com

## Abstract

Accelerating large language model pre-training is a critical issue in present research. In this paper, we focus on speeding up pre-training by progressively growing from a small Transformer structure to a large one. There are two main research problems associated with progressive growth: determining the optimal *growth schedule*, and designing efficient *growth operators*. In terms of *growth schedule*, the impact of each single dimension on a schedule's efficiency is under-explored by existing work. Regarding the *growth operators*, existing methods rely on the initialization of new weights to inherit knowledge, and achieve only non-strict function preservation, limiting further improvements on training dynamics. To address these issues, we propose Masked Structural Growth (MSG), including (i) growth schedules involving all possible dimensions and (ii) strictly function-preserving growth operators that is independent of the initialization of new weights. Experiments show that MSG is significantly faster than related work: we achieve up to **2.2x** speedup in pre-training different types of language models while maintaining comparable or better downstream performances.[1]

## 1 Introduction

Pre-trained language models (PLMs) (Devlin et al., 2019; Brown et al., 2020; OpenAI, 2023) have brought tremendous potential to research and applications. However, the excessive computational cost of pre-training is still a bottleneck. In addition to the rents on computational platforms, the delay in research cycles and increased carbon footprints (Schwartz et al., 2020) are also undeniable issues. While structured pruning (Wang et al., 2020; Xia et al., 2022; Chen et al., 2021) can expedite pre-training, the resultant models are smaller in size and can be inferior in performance and knowledge capacity (Li et al., 2020), limiting their adaptability. Instead, we study the problem of accelerating pre-training while still producing large-scale models. We focus on *progressively growing* from smaller models to larger ones, which is an intuitive idea inspired by the neurogenesis in human brain (Eriksson et al., 1998; Van Praag et al., 2002; Deng et al., 2010). Although research like the Chinchilla laws (Hoffmann et al., 2022) provides insights on the optimal model size given a FLOPs budget, it only considers fixed model sizes in training. Thus, the best practices for progressive growth remain a largely uncharted territory.

We consider two main problems: the *growth schedule* and the *growth operator*. *Growth schedule* determines when and where to grow the model structure. Existing work (Gong et al., 2019; Gu et al., 2021) has studied growth schedules for Transformers (Vaswani et al., 2017) involving layer number and the width of FeedForward Networks (FFNs). However, finding an efficient multi-staged schedule involving all the possible growth dimensions remains difficult. *Growth operator* stands for the operations applied during growth to inherit knowledge from the preceding model, for which function preservation (Chen et al., 2016; 2022) is a theoretically important property because it insures that

---

[*]Corresponding author

[1]Code is publicly available at https://github.com/cofe-ai/MSG.

Table 1: Summary of related work. ∘ stands for "supported but not function-preserving"; ✗ stands for "not supported"; □ stands for "supported, sometimes function-preserving, but can be problematic in other cases (Section 2.3.1, 2.3.2)"; ✓ stands for "supported and function-preserving in all cases".

| Method | layer_num | hidden_dim | ffn_dim | head_num | Any Initialization | Schedule | Speed-up Ratio* |
|---|---|---|---|---|---|---|---|
| *Stacking* (Gong et al., 2019) | ∘ | ✗ | ✗ | ✗ | ✗ | multi-stage | 1.65x |
| *CompoundGrow* (Gu et al., 2021) | ∘ | ✗ | □ | ✗ | ✗ | multi-stage | 1.82x |
| *Staged* (Shen et al., 2022) | □ | □ | □ | ✗ | ✗ | one/two-stage | ˜1.4x |
| *Bert2BERT* (Chen et al., 2022) | ∘ | □ | ✓ | ✓ | ✗ | one-stage | 1.82x |
| *LiGO* (Wang et al., 2022) | ∘ | ∘ | ∘ | ∘ | ∘ | one-stage | 1.82x |
| *MSG* (ours) | ✓ | ✓ | ✓ | ✓ | ✓ | multi-stage | 2.2x |

\* Most results in this column are not directly comparable because of the vastly different experimental settings among all the work. We just summarize the highest results reported under certain setting, and discuss the fair comparison topic in Section 4.2.

the initialized large model behaves the same as the small model. There are two potential drawbacks associated with existing operators: first, the function preservation is not strict, leading to function disparities in certain cases; second, they rely fully on the initialization of new weights, constraining improvements in training dynamics (Section 2.3.3).

To help address these issues, we propose Masked Structural Growth (MSG), a novel progressive learning framework for language model pre-training with Transformers. We leverage a *masking* mechanism to ensure function preservation by first eliminating the effects of the new neurons, and then gradually enhancing their roles in subsequent training. MSG offers growth operators for all possible dimensions (Section 2.2) with decent flexibility in schedule design, resulting in growth schedules with state-of-the-art speed-up ratios of **2.2x** for Bert (Devlin et al., 2019) and **1.4x** for GPT-2 (Radford et al., 2019) while maintaining comparable or better downstream performances. By solving a dilemma caused by Layer Normalization (Ba et al., 2016), MSG achieves *strict* function preservation in arbitrary expansion, for the first time. Moreover, MSG operators are independent of the specific initialization strategy of new weights, making it friendly to future study.

Progressive growth also proves feasible in training large language models (LLMs), but it is difficult to conduct systematic and rigorous studies due to limited resource and variety in settings. For related topics, please refer to the FLM-101B technical report (Li et al., 2023) which incorporates operators originated from MSG.

To summarize, our contributions include (i) we propose MSG: a novel framework for progressive pre-training, and develop growth schedules with state-of-the-art speed-up; (ii) we demonstrate that MSG is a solid backbone for future research due to its strict function preservation and independence of new weight initialization; (iii) experiments show that MSG outperforms existing methods across diverse model configurations, and provide insights on the role of function preservation.

## 2  PRELIMINARIES

We define the task in Section 2.1, describe the growth dimensions in Section 2.2, and discuss function preservation in Section 2.3 to explain the motivation of MSG.

### 2.1  TASK FORMULATION

Our objective is to find the optimal growth schedule $s = (s_0, ..., s_T) \in \mathcal{S}$ and operator $g = (g_0, ..., g_T) \in \mathcal{G}$ to minimize the total wall (real-world) time of pre-training a model $f$, keeping its downstream performance on dataset $\mathcal{D}$ intact. Each $s_t$ in $s$ is the intermediate structure after the $t$-th growth, represented by a hyperparameter set (Section 2.2); operator $g_t$ controls how the weights of $s_t$ is initialized to form a new model $f_t$ based on the previously trained model $f_{t-1}^*$:

$$f_t = g_t(f_{t-1}^*, s_{t-1}, s_t). \tag{1}$$

Let $c(f_t)$ be the wall time cost for training each $f_t$ to be $f_t^*$, and $\hat{f}$ be a model with the same structure as the final $f_T^*$ but trained from scratch without growing. Then, for each intermediate model, the ratio $c(f_t)/c(\hat{f})$ is measurable with analytical expressions like (Hoffmann et al., 2022). Finally, for some evaluation metric $L$ (the higher the better), our task is formulated as:

$$\min_{g \in \mathcal{G}, s \in \mathcal{S}} \sum_{t=0}^{T} c(f_t), \ s.t. \ L(f_T^*, \mathcal{D}) \geq L(\hat{f}, \mathcal{D}). \tag{2}$$

### 2.2  GROWTH DIMENSIONS OF TRANSFORMERS

There are 4 expandable dimensions in Transformers (Vaswani et al., 2017). We name them by the corresponding hyperparameters: $s_t = (hidden\_dim, ffn\_dim, head\_num, layer\_num)$. *hidden_dim* is the size of feature embedding; *ffn_dim* is the intermediate layer size of the FeedForward Networks

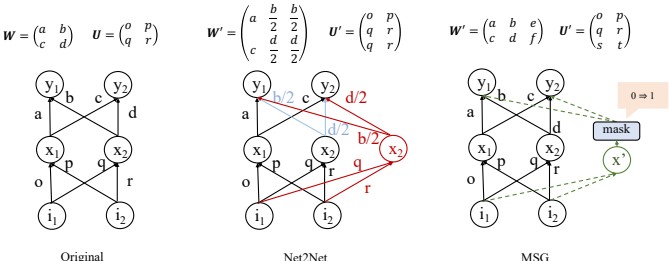

Figure 1: MSG (right) vs. Net2Net (middle) in the expansion of fully-connected layers.

(FFN); *head_num* is the number of heads in each Multi-Head Attention (MHA) module; *layer_num* is the total number of Transformer blocks. For instance, Bert-base is represented as (768, 3072, 12, 12) and Bert-large is (1024, 4096, 16, 24) (Devlin et al., 2019).

## 2.3 FUNCTION-PRESERVATION

In equation 1, if $g_t$ insures that

$$\forall x, f_t(x) = f_{t-1}^*(x), \tag{3}$$

we name $g_t$ a *function-preserving* operator. In this section, we first prove that most existing operators do not achieve strict function preservation in Layer Normalization (2.3.1) and the depth dimension (2.3.2). Next, we study the initialization for new weights and discuss in which cases could strict function preservation be critical (2.3.3). A summary of the mentioned methods is in Table 1.

### 2.3.1 THE LAYER NORMALIZATION DILEMMA

Layer Normalization (LN) (Ba et al., 2016) is formulated as:

$$\text{LN}(\boldsymbol{x}) = \boldsymbol{\omega} \odot (\boldsymbol{x} - \mu)/\sigma + \boldsymbol{\beta}, \tag{4}$$

where $\boldsymbol{x} \in \mathbb{R}^d$ is a feature vector, $\boldsymbol{\omega} \in \mathbb{R}^d$ and $\boldsymbol{\beta} \in \mathbb{R}^d$ are trainable weight and bias. In most implementations, $\mu \in \mathbb{R}$ and $\sigma \in \mathbb{R}$ are the mean and standard deviation of the $d$ elements in $\boldsymbol{x}$, respectively. $\odot$ denotes element-wise product. LN exists in a wide range of models. We prove that it raises a function-preserving dilemma for mainstream growth operators as follows:

**Net2Net** A widely-used operator for width expansion of fully-connected networks is Net2Net (Chen et al., 2016). We demonstrate it in Figure 1 (middle). While a layer $\boldsymbol{x}$ is growing from $n$ to $q$ neurons, Net2Net maps the $id$ of each new neuron $i$ to an existing neuron $m(i)$ following:

$$m(i) = \begin{cases} i & i \leq n \\ \text{map}(\{1, ..., n\}, i) & n < i \leq q \end{cases}. \tag{5}$$

Net2Net initializes new weights to ensure that neuron $i$ is a replica of $m(i)$ by copying from the input weights $\mathbf{U}$, and equally splitting the output weights $\mathbf{W}$:

$$\mathbf{U}'_{k,i} = \mathbf{U}_{k,m(i)}, \mathbf{W}'_{i,j} = \frac{1}{|\{x|m(x)=m(i)\}|}\mathbf{W}_{m(i),j}, \tag{6}$$

where $|\cdot|$ denotes cardinality. As a result, the output values of the consequent layers are preserved.

In Transformers, previous state-of-the-art methods (Chen et al., 2022; Qin et al., 2022b; Gu et al., 2021) are essentially different implementations of the mapping function (Eq. 5), and (Shen et al., 2022) holds a similar spirit. Unfortunately, Net2Net is not strictly function-preserving if LN is applied after $\boldsymbol{x}$. We take $\boldsymbol{x} = (x_1, x_2, x_3)$ for example: in a *hidden_dim* growth from 3 to 5, for the 2 new neurons, Net2Net selects existing neurons to copy. For $m(4)=2$ and $m(5)=1$, this yields:

$$\boldsymbol{x}' = (x_1, x_2, x_3, x_2, x_1) \Rightarrow \mu(\boldsymbol{x}) \neq \mu(\boldsymbol{x}') \Rightarrow \text{LN}(\boldsymbol{x}) \neq \text{LN}(\boldsymbol{x}'), \tag{7}$$

even if the LN $\omega$ and $\beta$'s are copied accordingly. Thus, $\boldsymbol{y}' = \mathbf{W}'(\text{LN}(\boldsymbol{x}')) \neq \boldsymbol{y}$, and the function preservation is broken. This holds regardless of the position of the activation functions (e.g., ReLU).

For weight-dependent methods like Net2Net, the only solution to this dilemma is *copying each neuron for exactly the same times* to preserve the mean and variance. However, this requires to at least double the width each time, resulting in a fixed exponential schedule that severely harms the flexibility. For example, a small model may have better training dynamics with $hidden\_dim = 512$ than 384 (Figure 2), but if our target model has $hidden\_dim = 768$, Net2Net must grow from a $384 = 768/2$ dimensional model to be function-preserving.

**Other Operators** Similarly, operators that zeroize the new input weights (Evci et al., 2022) yield $\boldsymbol{x}' = (x_1, x_2, x_3, 0, 0)$ and do not solve this dilemma. Moreover, they intrinsically require the activation function $\sigma$ to satisfy $\sigma(0) = 0$, which limits their usage. Thus, to our knowledge, there is no strictly function-preserving growth operator for arbitrary *hidden\_dim* growth before MSG.

### 2.3.2 THE DEPTH DIMENSION

For depth (*layer\_num*) growth, existing work either stacks (Gong et al., 2019; Yang et al., 2020; Chen et al., 2022; Gu et al., 2021) or inserts (Qin et al., 2022b; Li et al., 2022) new Transformer layers with weights copied from existing ones. Neither of them is function-preserving. Shen et al. (2022) zeroized all the LN weights in the new layer for GPT-like (Radford et al., 2019) Transformers:

$$\boldsymbol{x}' = \boldsymbol{x} + \text{Attention}(\text{LN}(\boldsymbol{x})), \quad \boldsymbol{y} = \boldsymbol{x}' + \text{FFN}(\text{LN}(\boldsymbol{x}')), \tag{8}$$

which achieved strict function preservation $\boldsymbol{y} = \boldsymbol{x}$. However, for Bert-like implementations:

$$\boldsymbol{x}' = \text{LN}(\boldsymbol{x} + \text{Attention}((\boldsymbol{x}))), \quad \boldsymbol{y} = \text{LN}(\boldsymbol{x}' + \text{FFN}(\boldsymbol{x}')), \tag{9}$$

this method yields all-zero output. Thus, a structure-agnostic function-preserving operator for depth growth is also underexplored.

### 2.3.3 DEPENDENCY ON INITIALIZATION

Most existing growth operators aiming at function preservation determine a *fixed* initialization for new weights. For example, Net2Net initializes new weights as a fraction/copy of existing weights, while other methods simply zeroize them. One of the drawbacks of these methods (especially the zeroizing ones) is the *Symmetry* issue: some new neurons always receive similar gradients to other new/existing neurons, which brings unnecessary restriction that slows down model convergence [2]. Furthermore, (Evci et al., 2022) and (Wang et al., 2022) show evidences that weight initialization can even be treated as an optimization problem itself. However, growth operators that depend on initialization deny this chance of further improving the training dynamics. Thus, although most work agree on the importance of function preservation, it seems necessary to answer "*it is worth sacrificing a better initialization?*" By MSG, we decouple these two factors and explore this topic by experimenting w/ and w/o function preservation in different conditions.

## 3 MASKED STRUCTURAL GROWTH

We propose a novel framework for progressive training, namely Masked Structural Growth (MSG). In Section 3.1, we introduce MSG operators for all growth dimensions. In Section 3.2, we discuss methodologies and results for schedule design.

### 3.1 THE MSG OPERATORS

The main idea of MSG is to derive external masks that completely eliminate the effects of new structures on the model's function. Immediately post-growth, with $mask = 0$, we make the function strictly preserved. Next, we gradually increase the masks to raise the influence of new structures, and finally achieve the target structure with $mask = 1$. The underlying rationales are (1) right after growth, the current loss precisely mirrors that of the smaller model, and it tends to reduce in the next step via gradient-descend; (2) when the mask value is close to 0, the outputs of existing neurons effectively provide initial guidance for the update of new weights.

### 3.1.1 FULLY-CONNECTED LAYERS

Fully-Connected (FC) layer is a linear projection between feature vectors, followed by (optional) activation and LN. It is involved in all the growth dimensions except *layer\_num*. Let $\boldsymbol{x} \in \mathbb{R}^{d_1}$ and

---

[2]Existing solutions to symmetry break the function preservation (Gong et al., 2019; Chen et al., 2022).

$y \in \mathbb{R}^{d_2}$ be the input and output of a FC layer $W$, respectively. As depicted in Figure 1 (right), while expanding the size of $y$ from $d_2$ to $d'_2$, we introduce a mask vector $c$ with $d_2$ elements of value 1 and $d'_2 - d_2$ elements of value 0:

$$c = [\mathbf{1}_{d_2}; \mathbf{0}_{d'_2 - d_2}], \tag{10}$$

where $[\cdot]$ stands for concatenation along the last dimension. We first copy the existing weights from $W$ and initialize the new weights to be $arbitrary$ value:

$$W'_{i,*} = \begin{cases} W_{i,*} & 1 \le i \le d_2 \\ \text{any value} & d_2 < i \le d'_2 \end{cases}. \tag{11}$$

Then, the new output $y' \in \mathbb{R}^{d'_2}$ of the grown FC layer is computed by:

$$y' = c \odot (\sigma(W' * x)) = [y; \mathbf{0}_{d'_2 - d_2}], \tag{12}$$

where $*$ stands for matrix (inner) product and $\sigma$ is any activation function. For existing neurons, the network yields exactly the same value as original; for new neurons, their outputs are masked to 0.

On the other hand, if the input $x$ is expanding to $x' \in \mathbb{R}^{d'_1}$, it should have been in the form of:

$$x' = [x; \mathbf{0}_{d'_1 - d_1}], \tag{13}$$

because it is the output of the previous layers and already processed by Equation (12). We initialize the new weights following:

$$W'_{*,j} = \begin{cases} W_{*,j} & 1 \le j \le d_1 \\ \text{any value} & d_1 < j \le d'_1 \end{cases}. \tag{14}$$

Thus, the output $y' \in \mathbb{R}^{d_2}$ of the network is:

$$y' = \sigma(W' * x') = \sigma(W * x) = y. \tag{15}$$

We do not need an additional mask here if the output size $d_2$ is not expanded. If both $d_1$ and $d_2$ are expanded, we first apply (14), then (11), and finally mask the output by (12). After solving LN (Section 3.1.2), the function of the whole FC module will be preserved, as illustrated in Figure 1.

### 3.1.2 LN SOLUTION

We solve the LN dilemma (Section 2.3.1) by incorporating a similar mask inside of LN. While expanding the size of a feature $x$ and its LN from $d$ to $d'$, we introduce an external mask $c \in \mathbb{R}^{d'}$:

$$c = [\mathbf{1}_d; \mathbf{0}_{d' - d}], \tag{16}$$

and derive a new formula for the mean and variance of the expanded feature $x' \in \mathbb{R}^{d'}$:

$$\begin{aligned} \mu' &= (x' * c)/\text{sum}(c), \\ (\sigma')^2 &= ((x' - \mu')^2 * c)/\text{sum}(c), \end{aligned} \tag{17}$$

where $\text{sum}(\cdot)$ stands for the sum of all elements and $*$ is inner product. With zero mask value, the new neurons has no contribution to the mean and variance. As a result, if we retain the first $d$ values of the LN weight $\omega$ and bias $\beta$, and initialize the $d' - d$ new LN weights to arbitrary value, we have:

$$\text{LN}^*(x') = [\text{LN}(x); \xi_{d'-d}], \tag{18}$$

in which the values of $\xi_{d'-d}$ are decided by the input weights connected to the new neurons in $x'$. Finally, we multiply mask $c$ again to the result:

$$c \odot \text{LN}^*(x') = [\text{LN}(x); \mathbf{0}_{d'-d}]. \tag{19}$$

Thus, for arbitrary $d$ and $d'$, MSG is strictly function-preserving with LN. As the training goes on, we gradually increase the last $d' - d$ values in $c$ to 1.

### 3.1.3 GROWTH OF SELF-ATTENTION HEADS

The self-attention module (Vaswani et al., 2017) contains multiple heads. Strict function-preservation can be achieved with MSG by treating each head as a "neuron" and adapting a similar process as equations 10 to 15. We leave the proof to Appendix A due to limited space.

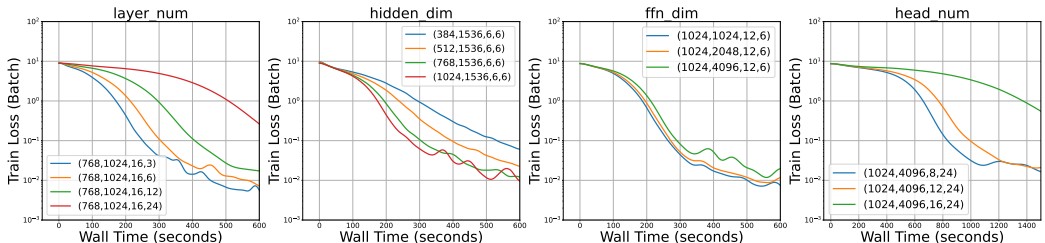

Figure 2: Training loss curves with different structural hyperparameters. We study the impact of each growth dimension on the model's "pre-training rate" $\gamma$ in early stages.

### 3.1.4 DEPTH GROWTH

We propose a depth growth operator inspired by residual connection (He et al., 2016). Let $\text{Layer}_n$ be a new Transformer layer, $\boldsymbol{x}_{n-1}$ be its input, and $\boldsymbol{c}$ be an all-zero mask in the same size as $\boldsymbol{x}_{n-1}$, we compute the layer output as:

$$\boldsymbol{x}_n = \boldsymbol{c} \odot \text{Layer}_n(\boldsymbol{x}_{n-1}) + (\boldsymbol{1} - \boldsymbol{c}) \odot \boldsymbol{x}_{n-1}. \tag{20}$$

With zero mask value, the whole $\text{Layer}_n$ is skipped to preserve function; while the mask value increases to 1, the residual connection vanishes, yielding a same structure as vanilla Transformers.

### 3.2 GROWTH SCHEDULE

In this sub-section, we study the construction of schedules. Although multi-staged growth inherently offers more flexibility than its single-staged counterpart, it is difficult to find an optimal growth schedule without being hindsight[3]. Existing work jump to the final schedules without detailed explanation Gong et al. (2019); Gu et al. (2021). In contrast, we introduce a methodology of *grid-searching*, which holds the potential to generalize to other Transformer variants. We do not claim to completely solve the problem, but provide empirical insights to inspire further investigations.

### 3.2.1 SCHEDULE CONSTRUCTION BY GRID SEARCH

An ideal growth schedule entails a series of structures that learn fast in their own stages. We find that a "pre-training rate" $\gamma$ is a reasonable indicator of learning speed in early/medium stages [4]:

$$\gamma \triangleq \frac{\Delta\text{loss}}{\Delta t} = \frac{\Delta\text{loss}}{\Delta\text{step}} \cdot \frac{\Delta\text{step}}{\Delta t} \triangleq \alpha \cdot \tau. \tag{21}$$

**Intermediate Structures.** Growth scheduling is a trade-off: a small model always has better constant $\tau$, but its $\alpha$ drops rapidly during training, making it necessary to grow in proper time. Given the challenge of quantifying $\alpha$ mathematically, we opt for a grid search with only 10k data to study how each of the 4 growth dimensions influences $\gamma$, respectively. We examine possible intermediate structures varying in different dimensions, and compare the training curves. The other hyperparameters are kept the same as in the final schedules. Models typically converge in several minutes with 8 GPUs, yielding minor time cost. As demonstrated in Figure 2, we find the following heuristic rules for Bert: (i) Large *layer_num* severely harms $\tau$ with limited gain in $\alpha$. It's preferable to grow it in late stages. (ii) Larger *hidden_dim* brings significantly better $\gamma$ for shallow models. It should start from a large value. (iii) The *ffn_dim* balances the $\alpha$ and $\tau$ itself with little impact on $\gamma$, while slightly smaller *head_num* brings better $\gamma$ for deep models. We construct different intermediate structure sequences based on these heuristics. The same methodology transfers to GPT.

**Stage Duration.** Like normal pre-training, the total training steps are pre-decided. Given the total steps and an intermediate structure sequence, we split the training process for each stage to have a roughly equal number of steps, while slightly emphasizing the early stages. We observe that this simple methodology is efficient in practice despite being theoretically sub-optimal. An optimal strategy may be growing to the next structure once the current $\gamma$ goes lower than the full target structure given the same wall time, which is obviously hindsight and not practical.

---

[3]Shen et al. (2022) studied this topic in a different setting, see Appendix B for more discussions.

[4]Note that $\gamma$ is no longer valid when the loss curve almost levels off. For example, in late stages, we observe no consistent correlation between training/validation loss and downstream performances for Bert.

Table 2: Growth schedules expanding one dimension in each stage. Sch1-B and sch1-L are used for Bert-base and Bert-large, and sch1-G for GPT-2, respectively. Rapid-L is a rapid schedule with multiple expansions within 100k steps.

| Schedule | 0~200k | 200k~400k | 400k~500k | 500k~600k | 600k~700k | 700k~900k |
|---|---|---|---|---|---|---|
| Sch1-B | (512,768,8,3) | (512,3072,8,3) | (512,3072,8,6) | (768,3072,8,6) | (768,3072,12,6) | (768,3072,12,12) |
| Sch1-L | (768,1024,12,6) | (768,4096,12,6) | (768,4096,12,12) | (1024,4096,12,12) | (1024,4096,16,12) | (1024,4096,16,24) |

| Schedule | 0~50k | 50k~100k | 100k~130k | 130k~160k | 160k~190k | 190k~end |
|---|---|---|---|---|---|---|
| Sch1-G | (512,768,8,3) | (512,3072,8,3) | (512,3072,8,6) | (768,3072,8,6) | (768,3072,12,6) | (768,3072,12,12) |

| Schedule | 0~400k | 400k~420k | 420k~440k | 440k~500k | 500k~800k |
|---|---|---|---|---|---|
| Rapid-L | (512,1024,8,6) | (512,4096,8,6) | (1024,4096,8,6) | (1024,4096,16,6) | (1024,4096,16,24) |

Table 3: Experiments on the mask strength $\phi$.

| $\phi$ | 0 | 100 | 250 | 500 | 800 | 1000 |
|---|---|---|---|---|---|---|
| loss@10k | 3.36 | 3.34 | 3.25 | 3.23 | 3.23 | 3.23 |

**Schedules Constructed.** Following the same methodologies mentioned above, we construct the schedules for both Bert and GPT, as detailed in Table 2. Sch1-B, Sch1-L, and Sch1-G are used in our main experiments for Bert-base, Bert-large, and GPT-2, respectively. These schedules are multi-staged with one dimension expanded per stage, leading to 1.8x, 2.2x, and 1.4x speed-up compared to training from scratch with the same step numbers. This is significantly faster than (Gong et al., 2019; Gu et al., 2021) on the same target models. Rapid-L is a rapid schedule with several successive growth stages arranged within a span of 100k steps to study the influence of stage duration. Other schedules for ablation study are explained in Appendix G.

**Mask Strength.** We define $\phi$ as the number of steps used to linearly increase the mask values from 0 to 1. Initially, based on intuition, we set $\phi = 5000$ for all our experiments, which worked well. To further study its impact, we experiment with different $\phi$ values on Bert-large with only 100k data. We train for more than 10 epochs to simulate the middle and late phases of the full-data training. The schedule we use here is a compressed version of Rapid-L (Table 2). We compare the training loss at 10k steps (Table 3) and observe that all $\phi$ values exceeding 500 can ensure stability in subsequent training and yield similar results.

### 3.2.2 INITIALIZATION

MSG supports arbitrary initialization of new weights. We find that sampling from a normal distribution $\mathcal{N}(0, 0.02)$ for width growth and following the *stacking* strategy for depth (complemented with eq. 20) already outperforms state-of-the-art methods. Given that $\mathcal{N}(0, 0.02)$ is a prevalent choice for initializing Transformers from scratch, we believe that it has been optimized through community-driven refinements. We leave the incorporation of other "learning to initialize" methods for future. For the optimizer, we copy the stored moving averages of AdamW (Loshchilov & Hutter, 2018) for existing weights, and start from scratch for new weights.

## 4 EXPERIMENTS

### 4.1 SETTINGS

We train both Bert-base and Bert-large (Devlin et al., 2019) using a pre-processed combination of English Wikipedia and Book Corpus (Zhu et al., 2015), and GPT-2 (Radford et al., 2019) using OpenWebText (Gokaslan & Cohen, 2019). The growth schedules are presented in Table 2. For other details, please see Appendix D. As for evaluation, we fine-tune our Bert models on the GLUE (Wang et al., 2018) and SQuADv1.1 (Rajpurkar et al., 2018) tasks, and GPT models on Wikitext2 (Merity et al., 2016). We report the mean and standard deviation of the metrics across 3 runs on the dev set. Detailed fine-tuning settings are provided in Appendix E.

### 4.2 COMPARED METHODS

In complement to Table 1, we answer the following two questions for more fair comparison:

*On which schedule should we compare?* If the growth operators ensure no performance drop, the speed-up ratio is fully dependent on the schedule. Since MSG succeeds on our own schedules which is faster than the previous state-of-the-art schedules (Section 3.2.1) for the same target models, we compare different operators on our own schedules.

Table 4: Main results and ablation studies on Bert-Large. We report the average GLUE score and the accuracy/F1 for SQuaD. The numbers are mean (standard deviation) computed across 3 runs.

| Target | Bert-Large | | | | | |
|---|---|---|---|---|---|---|
| Schedule | Full | Sch1-L | | | Rapid-L | |
| Wall time | 70h, 48min | **32h, 10min** | | | 29h, 20min | |
| Operator | - | Bert2BERT | MSG | No Mask | MSG | No Mask |
| Glue Avg. | 82.2(0.2) | 82.1(0.3) | **83.2**(0.2) | 82.2(0.2) | **79.1(0.4)** | 77.9(0.3) |
| SquaD | 80.2(0.2)/87.9(0.3) | 80.0(0.1)/87.6(0.1) | **81.3(0.2)/88.4(0.1)** | 79.7(0.3)/88.0(0.2) | **76.5(0.6)/84.4(0.5)** | 75.7(0.4)/84.0(0.3) |

Table 5: Main results and ablation studies on Bert-Base and GPT-2. For GPT-2, we evaluate on the validation set of Wikitext2 (Merity et al., 2016). "PPL-zs" stands for zero-shot perplexity (standard practice at the moment), while "PPL-ft" is fine-tuned with the Wikitext2 training data.

| Target | Bert-Base | | | Target | GPT-2 | | |
|---|---|---|---|---|---|---|---|
| Schedule | Full | Sch1-B | | Schedule | Full | Sch1-G | |
| Wall time | 26h, 10min | **14h, 36min** | | Wall time | 53h, 1min | **37h, 59min** | |
| Operator | - | Bert2BERT | MSG | Operator | - | Bert2BERT | MSG |
| Glue Avg. | 80.7(0.2) | 80.5(0.2) | **81.0(0.2)** | PPL-zs | 41.31 | 42.24 | **41.20** |
| SquaD | 79.1(0.2)/86.9(0.2) | 79.0(0.1)/86.7(0.0) | **79.6(0.5)/87.2(0.4)** | PPL-ft | **23.90** | 25.13 | 24.14 |

*What operators should we compare with?* Mainstream growth operators include zeroizing and Net2Net. Since zeroizing limits the usage of activation functions (Section 2.3.1) and sometimes lead to problematic training (Gong et al., 2019), most related work (Gong et al., 2019; Gu et al., 2021; Chen et al., 2022; Qin et al., 2022b) apply a combination of Net2Net on width and some "direct copy" strategy on depth (Table 1). Thus, for the main experiments, we compare with our reproduction of the latest version of the *Bert2BERT* (Qin et al., 2022b) operators that achieved state-of-the-art performances. It features the same Net2Net operators for width growth as (Chen et al., 2022), and an interpolation strategy for depth. Other configurations are kept the same as MSG. We compare with the data-driven *LiGO* method (Wang et al., 2022) in Appendix H due to limited space [5].

## 4.3 MAIN RESULTS

We present Bert-Large results in Table 4. Bert-base and GPT-2 results are in Table 5. The results on each sub-task is left to Appendix F due to limited space. We have the following observations:

*MSG schedules achieve higher speed-up.* Comparing with the no-grow baselines (Full-∗), MSG reaches equal downstream performances on Bert-base with 1.8x speed-up, and significantly higher performances on Bert-large with 2.2x speed-up. Comparing with the previous best multi-staged strategy (Gu et al., 2021) with 1.74x speed-up on Bert-base and 1.82x on Bert-large, our advantage is because MSG supports growth on all the 4 possible dimensions, which enables more flexible schedule design, while *CompoundGrow* (Gu et al., 2021) explored only 2 of them. The same holds for GPT-2 with 1.4x speed-up and comparable performances with baselines [6].

*MSG operators have better potential than Bert2BERT.* Comparing with Bert2BERT on the same schedules, MSG achieves significantly better performances on Bert-large and GPT-2. This is because (1) MSG is strictly function-preserving, saving the time for recovering function; (2) MSG supports random initialization, which naturally solves the *Symmetry* problem and leads to better dynamics.

## 4.4 ABLATION STUDY

We try to answer the following questions with ablation studies:

*Factors involved in schedule design.* Comparing the results of Rapid-L and Sch1-L (Table 4), even though the time cost is close, we observe significant performance gap. This indicates that rapid growth (expanding the model for too many times in a short period) is not friendly to Transformers. This supports the effectiveness of our intuitive strategy to equally split the stage duration (Section 3.2.1). Another factor may be the selection of intermediate structures. To study this, as an alternative schedule, we further develop the Sch0 (Appendix G and Table 8) following the same heuristics as

---

[5]The Gradmax (Evci et al., 2022) operators have not proven to be directly applicable to Transformers.

[6]For GPT-2, *Bert2BERT* starts from a large checkpoint with a *hidden_dim* of 512 and 12 layers without taking its training cost into account; *LiGO* reports 1.22x speed-up with a relatively small starting checkpoint. We observe that most methods have lower speed-up on GPTs than Bert and we guess this is because the causal language model loss decrease relatively slow for shallow models (3 layers, etc.).

Sch1 but with different intermediate structures. The GLUE results with Sch0 are presented in Table 9. The scores are close to Sch1 (82.8 vs. 83.2 on Bert-Large, 80.9 vs. 81.0 on Bert-base), supporting the robustness of our schedule-construction methodologies.

*Importance of function preservation.* To study the effect of function preservation, we remove all the masks right after growth (*No Mask* in Table 4), resulting in a strategy of randomly initializing new weights without function preservation. Although (Chen et al., 2022) showed that a similar strategy is inferior in one-staged settings, we observe that it performs reasonably well on our Sch1-L, comparable to both the Full baseline and Bert2BERT (Table 4). This indicates that random initialization can indeed result in better training dynamics than the weight-dependent Bert2BERT. On the other hand, *No Mask* underperforms MSG on both Sch1-L and Rapid-L with the same initialization strategy, which supports that function preservation is beneficial if it does not harm the initialization. Moreover, looking at the training loss curve of *No Mask* vs. MSG on Rapid-L (Figure 3), we observe that MSG will be more stable than *No Mask* if the training continues. This indicates that for rapid structural changes and limited time budget, function preservation improves the training stability.

## 5 RELATED WORK

### 5.1 ACCELERATING PRE-TRAINING

Progressive training (Wen et al., 2020; Dong et al., 2020) has raised attention for its ability to accelerate pre-training for both Computer Vision (Li et al., 2022) and Natural Language Processing (Gong et al., 2019; Gu et al., 2021). Gong et al. (2019) proposed a *stacking* method which doubles the model depth in each stage. *CompoundGrow* (Gu et al., 2021) extends *stacking* by adding FeedForward Network (FFN) expansion into schedule. Shen et al. (2022) proposed a *staged* method that further supports expanding the hidden size of features, but limited to integer multiplications in which the feature size, FFN size, and head number are bounded. *Bert2BERT* (Chen et al., 2022; Qin et al., 2022b) and *LiGO* (Wang et al., 2022) support all possible growth dimensions, but focus on one-stage growth where all the dimensions are expanded simultaneously. However, each growth dimension has different impact on the training dynamics (Section 3.2), which is underexplored. In contrast, we study multi-staged growth with only one dimension expanded each time, resulting in higher degree of flexibility and speed-up. A detailed summary is in Table 1. Other work that speeds up pre-training include partial update (Yang et al., 2020), weight sharing (Lan et al., 2019), adversarial training (Clark et al., 2019), compute-optimal scaling laws (Hoffmann et al., 2022), and Mixture of Experts (MoEs) (Fedus et al., 2022), which are orthogonal to our topic. These approaches are systematically surveyed by (Bartoldson et al., 2023).

### 5.2 FUNCTION-PRESERVING GROWTH

*Function-preserving* growth ensures that post-growth models mirror the outputs of their precursors for any input, which is intuitively beneficial to knowledge inheritance. It is typically tackled mathematically to avoid conducting the time-consuming data-driven distillation (Jiao et al., 2020; Qin et al., 2022a). Chen et al. (2016) proposed the Net2Net method that preserves function in width expansion by splitting existing weights to form new neurons. Other methods (Evci et al., 2022; Wei et al., 2016) preserve function by setting some weights to zero. While applied to Transformers structure, existing operators can only achieve *non-strict* function preservation in most cases, limiting their efficacy (Section 2.3). Besides, they fully rely on the initialization of new weights. In contrast, our method is strictly function-preserving in all growth dimensions and is agnostic to new weight initialization. *LiGO* (Wang et al., 2022) designed an efficient data-driven method to map the weights between small and large models, which improves the training dynamics by learning to initialize. Due to MSG's unique advantage of supporting arbitrary initialization, it can be combined with *LiGO*-like methods on the new weights. We leave this interesting topic for future.

## 6 CONCLUSION

We proposed a novel framework: Masked Structural Growth (MSG) for progressive pre-training of language models. MSG supports growth operations on all the possible dimensions for efficient schedules. Experimental results show that MSG achieves state-of-the-art speed-up on Bert-base, Bert-large, and GPT-2 with equal or improved performances on downstream tasks. We also prove that MSG is strictly function-preserving as well as independent of new weight initialization. These properties make MSG outperform other operators and benefit future research.

LIMITATIONS

In this work, we have proposed the heuristic rules for growth schedules involving one dimension at a time. However, there is still unexplored space because subsets of several growth dimensions can be combined in the same stage. We believe that there is still a long way to go for *self-adaptive* growth schedules that actively search for the next target structures base on training dynamics. Another limitation of our work is that the best initialization strategy for Transformer growth remains an open question. Although we have showed that random initialization can be better than Net2Net, we leave more in-depth study (i.e. a generalization of (Evci et al., 2022) to Transformers) for future work.

ACKNOWLEDGMENTS

This work was supported by the National Science and Technology Major Project (2022ZD0116300) and the National Science Foundation of China (No. 62106249).

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

## A    PROOF OF MSG FUNCTION-PRESERVATION ON ATTENTION HEADS

In previous work, the size $d$ of query, key and value vectors in each head is decided by *hidden_dim* / *head_num*. In this work, we break this binding by fixing $d = 64$ and expanding the self-attention module via adding new heads. Weight matrices $\boldsymbol{Q}, \boldsymbol{K}, \boldsymbol{V}$ are all in size (*hidden_dim*, $d$). While growing from $n_1$ to $n_2$ heads, we keep the weights for existing heads and initialize the new heads arbitrarily. For head $i$:

$$\boldsymbol{Q}'_i, \boldsymbol{K}'_i, \boldsymbol{V}'_i = \begin{cases} \boldsymbol{Q}_i, \boldsymbol{K}_i, \boldsymbol{V}_i & i \leq n_1 \\ \text{any value} & i \in (n_1, n_2] \end{cases}. \tag{22}$$

For all the new heads $n_1 < i \leq n_2$, we multiply an all-zero mask $\boldsymbol{c}$ to its value vector:

$$\boldsymbol{v}'_i = \boldsymbol{c} \odot (\boldsymbol{V}'_i * \boldsymbol{x}) = \boldsymbol{0}, n_1 < i \leq n_2, \tag{23}$$

and get all-0 attention outputs for $n_1 < i \leq n_2$:

$$\boldsymbol{o}'_i = \text{softmax}\left(\boldsymbol{q}'_i * (\boldsymbol{k}'_i)^T / \sqrt{d}\right) * \boldsymbol{v}'_i = \boldsymbol{0}, \tag{24}$$

where $\boldsymbol{q}'_i$ and $\boldsymbol{k}'_i$ contains all the queries and keys for an input sequence.

For the FC layer $\boldsymbol{W}_o \in \mathcal{R}^{\text{hidden\_dim} \times (d \cdot n)}$ that collects head outputs, we initialize it as:

$$(\boldsymbol{W}_o)'_{*,j} = \begin{cases} (\boldsymbol{W}_o)_{*,j} & j \leq d \cdot n_1 \\ \text{any value} & j \in (d \cdot n_1, d \cdot n_2] \end{cases}. \tag{25}$$

Finally, the self-attention output is function-preserving for any $n_1$ and $n_2$:

$$\boldsymbol{y}' = \boldsymbol{W}'_o * [\boldsymbol{o}'_1; ...; \boldsymbol{o}'_{n_2}] = \boldsymbol{W}_o * [\boldsymbol{o}_1; ...; \boldsymbol{o}_{n_1}] = \boldsymbol{y}. \tag{26}$$

## B    DIFFERENT RESEARCH PROBLEMS IN SCHEDULES

Research on *staged* training (Shen et al., 2022) provided insightful analysis on the problem of automatically deciding the time (step) to grow. They perform similar experiments as our Section 3.2.1 to find constants related to training curves that help this decision, although still relying on hindsights.

The main difference is that they actually answer the question of "*given the structure sequence, on which step should we grow?*", while our work in Section 3.2.1 focuses on answering "*how to design a good structure sequence, which dimension should grow first?*".

In their work, they solve our question by manually design the schedules, as most existing work did; in our work, we solve their question by assigning roughly equal steps for each stage. We believe that both are important topics for future research.

## C    AN INTRODUCTION TO BERT2BERT

Bert2BERT (Chen et al., 2022) is the most typical related work that incorporates Net2Net for width growth and a weight-copying strategy for depth growth. Specifically, the expansion of *hidden_dim* and *ffn_dim* are standard Net2Net applications on fully connected layers (Eq. 5 and 6). For *head_num*, they also considered each head as a neuron-like unit. The Net2Net mapping function for attention heads is written as:

$$m(i) = \begin{cases} i & i \leq a^s \\ \text{map}(\{1, ..., n\}, i) & a^s < i \leq a^t \end{cases}, \tag{27}$$

where $a^s$ and $a^t$ are the head numbers of the source and target models, respectively. The parameters in head $m(i)$ are direct copied from head $i$. In the depth dimension, they follow (Gong et al., 2019) to stack the whole model (except embedding layers) on itself repeatedly until the target *layer_num* is reached. In a following implementation (Qin et al., 2022b), an interpolation strategy is used instead, which copies each layer separately and puts the duplicate above the original layer. In Bert2BERT, a warmup stage is introduced before the main pre-training stage, in which only the top layers are tuned. Note that MSG does not have this warmup stage due to its multi-staged nature.

Table 6: GLUE task results for Bert-Large.

| Schedule | Operator | GLUE Avg. | CoLA | SST-2 | MRPC | STS-B |
|---|---|---|---|---|---|---|
| Full | - | 82.2(0.2) | 57.9(1.3) | **91.9(0.1)** | 85.8(0.5)/89.7(0.2) | 89.0(0.1)/88.7(0.1) |
| Sch1-L | Net2Net | 82.1(0.3) | 57.6(1.3) | 91.9(0.2) | 85.1(0.8)/89.6(0.4) | 89.5(0.2)/89.0(0.2) |
| Sch1-L | MSG | **83.2(0.2)** | **61.9(0.4)** | 91.6(0.3) | **87.3(0.7)/91.0(0.4)** | **89.7(0.0)/89.2(0.0)** |
| Sch1-L | No Mask | 82.2(0.2) | 56.8(1.0) | 91.0(0.4) | 86.2(0.6)/90.4(0.3) | 89.1(0.1)/88.6(0.2) |
| Rapid-L | MSG | **79.1(0.4)** | **48.0(1.8)** | **90.7(0.4)** | **81.3(0.6)/87.3(0.3)** | **87.4(0.2)/87.0(0.1)** |
| Rapid-L | No Mask | 77.9(0.3) | 45.0(2.0) | 88.6(0.2) | 77.3(1.0)/85.0(0.6) | 86.7(0.1)/86.3(0.1) |

| Schedule | Operator | QQP | MNLI(m/mm) | QNLI | RTE | SQuADv1.1 |
|---|---|---|---|---|---|---|
| Full | - | 90.7(0.3)/87.7(0.4) | 83.3(0.5)/83.6(0.3) | 90.4(0.2) | 68.1(0.7) | 80.2(0.2)/87.9(0.3) |
| Sch1-L | Net2Net | 90.8(0.2)/87.8(0.3) | 82.9(0.3)/83.2(0.2) | 90.4(0.2) | 68.0(1.1) | 80.0(0.1)/87.6(0.1) |
| Sch1-L | MSG | **90.9(0.1)/87.9(0.2)** | **83.7(0.5)/84.5(0.3)** | 91.2(0.3) | **69.0(1.6)** | **81.3(0.2)/88.4(0.1)** |
| Sch1-L | No Mask | 90.8(0.2)/87.7(0.2) | 83.7(0.3)/84.0(0.4) | **91.3(0.3)** | 68.5(1.1) | 79.7(0.3)/88.0(0.2) |
| Rapid-L | MSG | 89.8(0.4)/86.2(0.6) | **80.1(0.1)/80.8(0.2)** | 88.2(0.2) | 66.1(0.7) | **76.5(0.6)/84.4(0.5)** |
| Rapid-L | No Mask | **89.8(0.1)/86.4(0.2)** | 79.7(0.2)/79.7(0.2) | 87.3(0.1) | **66.7(0.3)** | 75.7(0.4)/84.0(0.3) |

## D  DETAILS FOR PRE-TRAINING

**Bert**  Bert-base has 110M parameters, while Bert-large has 336M parameters. For the baseline models without growth (Full-*), we train for 1M steps, since the downstream performances continued to increase between 900k to 1M steps. We use a learning rate of (1e-4,1e-5) and warm-up step of (10k,30k) for (Bert-base, Bert-large), respectively, with a linear learning rate scheduler for both. The learning rate is reset to its maximum value at the 1st and 2nd growth stage, following (Gu et al., 2021). The batch size is set to 256 and maximum sequence length to 128. We clip the gradient norm to 1.0 for Bert-large. All the experiments are conducted on a single pod with 8 Nvidia A100 GPUs. The pre-training dataset includes 67 million sentence pairs with 15% tokens masked.

**GPT**  GPT-2 has 110M parameters. We train with a sequence length of 1024 and a batch size of 140 for 240k steps, which is 2 epochs on OpenWebText. For schedule Sch1-G, we continue training with the full model size until reaching the time cost reported in Table 5. We use a learning rate of 1e-4 and a learning rate warmup ratio of 0.01. The experiments are conducted on 2 pods with 14 Nvidia A100 GPUs.

## E  DETAILS FOR FINE-TUNING

For all the GLUE tasks, we use a batch size of 32, sequence length of 128 and learning rate of 2e-5. We fine-tune for 5 epochs for small datasets including CoLA, MRPC, STS-B, and RTE, (we exclude WNLI following most related work (Devlin et al., 2019; Chen et al., 2022; Gong et al., 2019)), and 3 epochs for other tasks.

For SQuAD, we fine-tune with a batch size of 12 and learning rate of 3e-5 for 2 epochs. SQuAD metrics are very sensitive to sequence length, and most models work well only with more than 384 sequence length. Thus, we continue pre-training after the whole schedule with a sequence length of 512 for 100k steps for all the methods compared. This yields a slight drop in actual speed-up ratios from 2.2x to 2.0x on Bert-large. Obviously, this drop can be alleviated if a sequence length of 512 is used through the whole pre-training process. Moreover, the SQuAD results of MSG are significantly higher compared to the baselines in all cases (Table 4, 5), which can be converted to additional time savings.

For Wikitext2 fine-tuning, we use a learning rate of 1e-4 for 3 epochs with a sequence length of 1024 and a batch size of 8. We report the zero-shot and fine-tuned perplexities on the validation set.

## F  RESULTS ON SPECIFIC GLUE TASKS

We present the scores on each GLUE task in Table 6 and 7. For metrics, we use Matthews correlation for CoLA, Pearson/Spearman correlation for STS-B, accuracy/f1 for MRPC, QQP, and SQuAD, and accuracy for all the others. The numbers are mean (standard deviation) computed across 3 runs.

Table 7: GLUE task results for Bert-Base.

| Schedule | Operator | GLUE Avg. | CoLA | SST-2 | MRPC | STS-B |
|----------|----------|-----------|------|-------|------|-------|
| Full | - | 80.7(0.2) | 52.2(0.8) | 90.4(0.2) | **85.9(0.9)/90.1(0.5)** | **88.8(0.1)/88.4(0.1)** |
| Sch1-B | Net2Net | 80.5(0.2) | 52.4(1.5) | **91.1(0.4)** | 84.3(0.9)/88.8(0.6) | 88.3(0.1)/88.0(0.1) |
| Sch1-B | MSG | **81.0(0.2)** | **58.2(1.6)** | 91.0(0.2) | 85.0(0.5)/89.4(0.5) | 88.1(0.1)/87.6(0.1) |

| Schedule | Operator | QQP | MNLI(m/mm) | QNLI | RTE | SQuADv1.1 |
|----------|----------|-----|------------|------|-----|-----------|
| Full | - | **90.6(0.1)/87.3(0.1)** | **82.5(0.3)/82.9(0.1)** | 89.9(0.1) | **65.1(0.7)** | 79.1(0.2)/86.9(0.2) |
| Sch1-B | Net2Net | 90.1(0.3)/87.0(0.1) | 81.1(0.2)/82.1(0.1) | 89.2(0.1) | 66.3(0.5) | 79.0(0.1)/86.7(0.0) |
| Sch1-B | MSG | 90.0(0.1)/87.0(0.1) | 81.8(0.3)/82.4(0.2) | **89.9(0.1)** | 63.1(1.6) | **79.6(0.5)/87.2(0.4)** |

Figure 3: Training loss curves of MSG with ans without mask on the Rapid-L schedule.

## G  ALTERNATIVE SCHEDULES AND RESULTS

According to our heuristic rules in Section 3.2.1, starting from maximum *head_num* and *hidden_dim* and growing only the other dimensions can be an alternative schedule. For ablation study, we construct such schedules, namely Sch0-L/B for Bert-large/base, as presented in Table 8. The GLUE results with Sch0 are shown in Table 9.

Table 8: Additional growth schedules. "B" and "L" stand for base and large model size, respectively.

| Schedule | 0~200k | 200k~400k | 400k~600k | 600k~900k | |
|----------|--------|-----------|-----------|-----------|---|
| sch0-L | (768,1024,16,6) | (768,4096,16,6) | (768,4096,16,12) | (768,4096,16,24) | |

| Schedule | 0~200k | 200k~400k | 400k~600k | 600k~700k | 700k~900k |
|----------|--------|-----------|-----------|-----------|-----------|
| sch0-B | (768,768,12,3) | (768,768,12,6) | (768,768,12,9) | (768,768,12,12) | (768,3072,12,12) |

## H  COMPARING WITH LIGO

We take the reported wall-time speedup numbers of *LiGO* from the Figure 2 in (Wang et al., 2022) for comparison because the training settings are very close to ours. The speedup of **MSG** vs. *LiGO* w.r.t. baseline models trained from scratch is **1.8x** vs. 1.69x on Bert-base, and **2.2x** vs. 1.82x on Bert-large. This indicates that both our mathematically strict operators and their data-driven operators can be effective for growth. Again, note that they are partially orthogonal (Section 5.2) because MSG is potentially compatible with arbitrary new weight initialization.

## I  SANITY CHECK: SCALING TO 100 BILLION PARAMETERS

MSG-oriented growth operators have been applied to train an LLM by growing from 16B to 51B, and then to 101B parameters (Li et al., 2023). Since a head-to-head comparison to full-size training is impossible given the computational budgets, we hereby summarize some sanity-check studies in

Table 9: Additional results on GLUE with MSG and Sch0-L/B.

| Schedule | Operator | GLUE Avg. | CoLA | SST-2 | MRPC | |
|---|---|---|---|---|---|---|
| Sch0-L (36h, 8min) | MSG | 82.8(0.1) | 60.0(0.8) | 91.9(0.2) | 86.8(0.7)/90.7(0.5) | |
| Sch0-B (17h, 46min) | MSG | 80.9(0.1) | 55.0(0.2) | 90.7(0.1) | 84.3(0.7)/88.9(0.5) | |
| Schedule | Operator | STS-B | QQP | MNLI(m/mm) | QNLI | RTE |
| Sch0-L (36h, 8min) | MSG | 88.9(0.5)/88.6(0.4) | 90.9(0.3)/87.9(0.2) | 83.7(0.3)/84.6(0.6) | 91.2(0.3) | 68.1(0.6) |
| Sch0-B (17h, 46min) | MSG | 88.6(0.2)/88.1(0.2) | 90.3(0.1)/87.1(0.0) | 82.4(0.2)/82.8(0.1) | 90.2(0.0) | 65.5(1.2) |

that technical report to show evidences that the training dynamics of MSG are stable for models at 100B+ scales, and the growth strategy successfully leverages the capacities of large models.

**Settings.** A GPT-like LLM is trained with 24 DGX-A800 GPU (8×80G) servers. The training data is composed of around 300B Chinese-English mixed tokens. Due to limited budgets, (Li et al., 2023) does not follow the methodologies in Section 3.2.1. Instead, the 16B, 51B, and 101B models are trained with 245.37B, 39.64B, and 26.54B tokens, respectively. This is a very "rapid" growth schedule. The target model hyperparameter is (10240, 40960, 80, 80). The training is completed in 21.54 days with a budget of $100K.

**Results.** As depicted in Figure 4 (picked from Figure 2 in (Li et al., 2023)), the training loss curve of MSG is stable in general after each growth, and the loss reduction rate w.r.t. number of tokens gets larger after growth. Besides, according to the evaluation results in their Table 5, the 101B model yields greater improvements than 50B with less data consumed. This indicates that the new weights are successfully incorporated in the training after MSG growth, which is consistent with the loss curve. The benchmark evaluations show that comparing with 100B-scale models trained with a similar amount of data (i.e. GLM-130B), the model achieves at least 80% of the baselines' performances with 10% of their training cost. Note that this is not a strict comparison since both the models and pre-training data are vastly different. To summarize, MSG-oriented operators show abilities to stabilize the post-growth training and inherit knowledge from small models in progressive training of 100B-scale models.

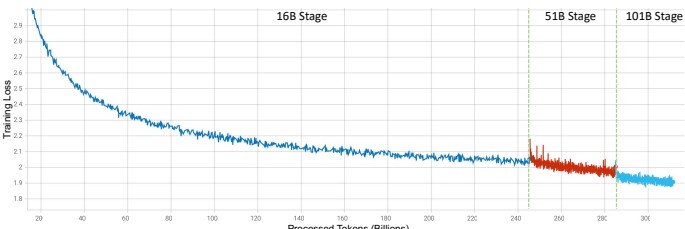

Figure 4: Training curve of the 101B model.

## J ADDITIONAL GPT RESULTS

As our GPT ppls on Wikitext2 in Table 5 are larger than those reported in the original GPT-2 work (Radford et al., 2019), we doubt that our settings in Appendix D led to insufficient pre-training. Thus, we tried a new setting: we increase the batch size to 480 and train for 100k steps in total, which is around 6 epochs of OpenWebText. The data processing is kept the same as a public implementation (NanoGPT[7]). This new setting results in a training loss of ~3.10, which is also close to the value reported by NanoGPT.

We develop a new schedule (Table 10) by our grid-searching methodology (Section 3.2.1). This schedule reduces the training time to 39h, 30min, comparing to 54h, 33min for full-size training, which is a 38% speedup. The training loss curves w.r.t steps are presented in Figure 5. The Wikitext2 ppls are 37.65 (MSG) vs 37.31 (Full), which are comparable. However, there is still a significant gap compared to (Radford et al., 2019) (~30). A possible explanation is that the open-sourced OpenWeb-Text data is different from the close-sourced ones used by GPT-2. Note that this does not influence our conclusions since all our methods are compared with the same data.

---

[7] https://github.com/karpathy/nanoGPT

Table 10: Schedule for additional GPT results.

| Schedule | 0~30k | 30k~60k | 60k~80k | 80k~100k |
|---|---|---|---|---|
| Sch-GPT-New | (512,768,8,12) | (768,768,8,12) | (768,768,12,12) | (768,3072,12,12) |

## K  MSG LOSS CURVES

The MSG training loss curves vs. steps are presented in Figure 5. The curves for GPT-2 are based on the settings in Appendix J, while those for Bert are from the main experiments (Section 4.3). We observe that MSG gradually catches up in training loss with progressive growth. We also observe that lower loss in late training does not necessarily indicate superior performances (for the full corpus is process for more than 1 epoch), and thus the loss curves are just a qualitative indicator of the models' capabilities.

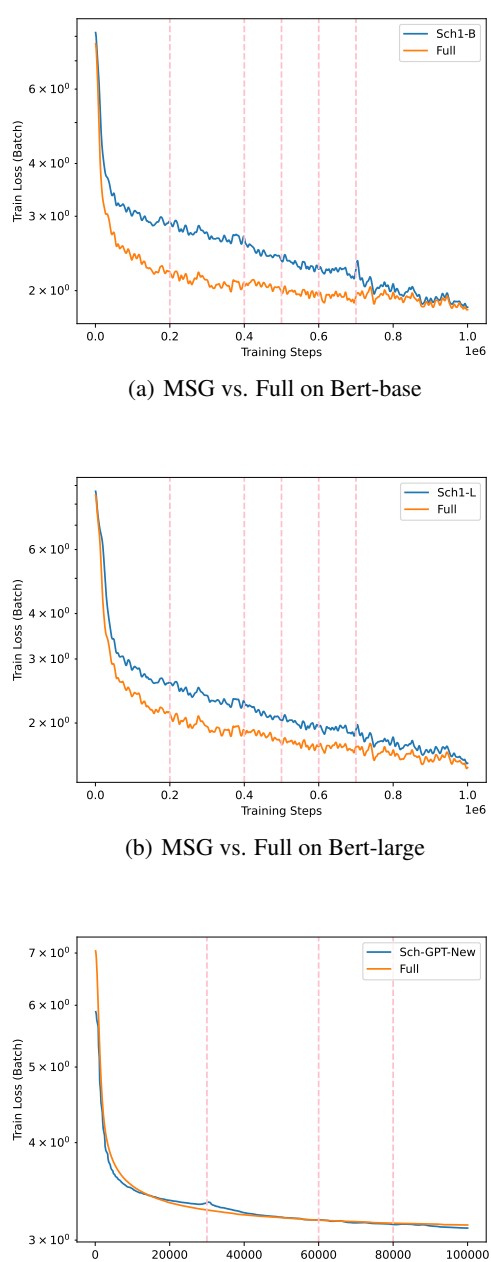

(a) MSG vs. Full on Bert-base

(b) MSG vs. Full on Bert-large

(c) MSG vs. Full on GPT-2

Figure 5: MSG loss curves. The pink vertical lines mark the beginning of each growth stage.

