# OpenReview forum: "Masked Structural Growth for 2x Faster Language Model Pre-training"
_ICLR.cc/2024/Conference — ICLR 2024 poster_

### Official Review · Reviewer_aWRp · 2023-11-01

**Soundness:** 3 good
**Presentation:** 3 good
**Contribution:** 3 good
**Rating:** 8
**Confidence:** 4

**Summary:**

The authors propose a framework called Masked Structural Growth (MSG) for progressive pre-training of transformer-based language models. The MSG includes the following two parts:
(1) growth schedules involving all the four possible dimensions;
(2) strictly function-preserving growth operators that are independent of the initialization of new weights.

Their experimental results show that the MSG achieves state-of-the-art speed-up on Bert-base, Bert-large, and GPT-2 with equal or improved performances on downstream tasks.

**Strengths:**

The greatest strength of the paper is that it introduces the MSG operators for all growth dimensions by using external masks.
These growth operators are strictly function-preserving (even with layer normalization), and independent of the initialization of new weights.
This apparently has the advantages over the growth operators proposed in the previous work.

Additionally, the growth schedule proposed in the MSG uses grid-search to grow only one dimension each time, which provides empirical insights on the growing models and holds the potential for generalization.

**Weaknesses:**

One weakness of the paper is that the authors have not empirically shown the advantages of the MSG in today's large-scale transformer models (which is much larger than the scale of BERT-large and GPT2). Such a weakness may be due to the lack of the computational resources.
In terms of presentation, the authors may want to add more description on the bert2BERT method, since it is one main competitor to the proposed MSG.

Some minor issues will be listed in the "Questions" section.

**Questions:**

I list my questions and some minor issues as follows:
(1).
For on page 4 and page 5:
Are vectors x, y, c in your section 3.1.1 all column vectors? If yes, then in Eq. (12),
x^T (the transpose of x) is not necessary, x will be good enough.

(2).
For Eq. (11) on page 5:
For d_2 < i < d_2', it should be:  d_2 < i <= d_2'

(3). Similarly,
For Eq. (14) on page 5:
For d_1 < j < d_1', should be: d_1 < j <= d_1'

(4).
Last line on page 7:
The stated (Qin et al., 2022b) is not the reference for the bert2BERT method.
Do you really mean the bert2BERT or the ELLE here?

---

> ### Author Response · Authors · 2023-11-20
>
> Thanks for your constructive suggestions!
> - Application to larger models: please see the common response.
> - We have added introductions to bert2BERT in Appendix (C) in the new version.
> - Equation 11, 12, 14 are fixed. We appreciate your careful reading.
> - ELLE reported an evolved version of bert2BERT which is applied to their own tasks. The methods are very similar with a main difference being an interpolation (instead of stacking) strategy in depth growth. We mean the implementation from ELLE here.
>
> We hope that this solves some of your concerns and we are more than happy to have further discussions.

---

### Official Review · Reviewer_vhLH · 2023-11-01

**Soundness:** 3 good
**Presentation:** 3 good
**Contribution:** 3 good
**Rating:** 6
**Confidence:** 3

**Summary:**

The manuscript focuses on training transformers faster by progressively growing the architecture during training. Problems with previous methods in this area include limited growth "dimensions" (e.g. inability to progressively grow the hidden dimension), absence of function preservation in networks with LayerNorm, and a forced choice between function preservation and initialization flexibility. The proposed method (MSG) addresses these problems, and experiments support its effectiveness (i.e., it accelerates training). Ablation and hyperparameter studies further justify the chosen methodology and offer insights that are helpful for understanding the progressive growth problem more broadly.

**Strengths:**

The proposed progressive growth method speeds up training more than competing approaches, and it maintains or improves accuracy/perplexity performances on test datasets (relative to the model without progressive growth).

The study of the impact of different growth dimensions is broadly helpful and clear (Figure 2). This analysis provides intuition for the MSG approach, making it seem less tailored to perform well on just one dataset/architecture (i.e., more generally applicable).

Care is taken to ensure that comparisons across progressive growth methods are fair and that the baseline progressive growth methods are relevant. In Section 4.2, salient features of progressive growth algorithms are isolated appropriately to facilitate analyses of the proposed method, MSG.

The manuscript offers an original and well-designed approach to disentangling the function preservation and initialization decisions of progressive learning. This facilitates the creation of a more flexible growth algorithm and better analysis of the importance of these algorithm components to progressive learning. For example, the final paragraph of Section 4.4 leverages this disentangling to show that both function preservation and flexible initializations matter to performance, supporting the form of the MSG method.

**Weaknesses:**

While the manuscript makes careful comparisons to competing progressive growth methods, some of the baseline performances (without progressive growth) are a little lower than expected. Please see my comments about this in the "Questions" section below.

**Questions:**

Score-affecting:

1. Please ensure all baseline approaches are faithful so that readers know MSG is effective in high-performing training runs. Examples follow.
   - In Tables 4 and 5, the SQuAD scores are 2-3% lower than what is reported in the BERT paper.
   - In Table 5, GPT-2 perplexity on Wikitext2 is high (41.3). There are strong public GPT-2 implementations; e.g., https://github.com/karpathy/nanoGPT.
      - It would be nice to see an MSG version of the LiGO (Wang et al., 2022) paper's Figure 3C: i.e., show perplexity/loss dynamics of GPT trained with and without MSG.

Helpful:

1. Please summarize the MSG operator conclusions/discussion from the referenced technical report (Li et al., 2023).
2. Consider moving tables and figures to the page they are discussed on, or to the next page. This helps readability.
3. Clean up figures and tables. Some example suggestions follow:
   - Table 2 could have uniform line lengths.
   - Figure 2 could have larger axis text, lines, legend text, etc.


Minor:

1. The manuscript would benefit from a proofreading. Some examples follow:
   - "Proof" is not a verb that means "to mathematically demonstrate". Say "we prove" instead of "we proof". A "proof" is what proves something.
   - On page 6, rephrase the second sentence of the "Stage Duration" paragraph -- the training steps are what gets split, not the stage steps.
2. I think that not all the inequalities in the second lines of equations 11 and 14 are strict (i.e., change a "<" to a "<=").
3. All of the methods listed in the Related Work section (progressive growth, weight sharing, MoEs, etc.) are discussed in the survey "Compute Efficient Deep Learning" (Bartoldson et al., 2023).

---

> ### Author Response · Authors · 2023-11-20
>
> Thanks for your careful reading and constructive suggestions! We believe that your suggestions help us a lot in improving the paper.
> - We tried our best with our current computational resources to study the performance gap problem you mentioned. Specifically, we first doubted that the high perplexity of our GPT model was because the model was undertrained. Thus, we controlled the training data to be exactly the same with nanoGPT, which is 9B OpenWebText tokens from huggingface. We performed a more sufficient training with a similar setting to nanoGPT and achieved very close training loss to the one they reported (3.11). Thus, we believe that our implementation is close enough to strong public repositories with this specific dataset, which makes our experiments meaningful. The settings and results are detailed in Appendix (J) in the new version. However, although we reduced the ppl from 41 to 37, there is still a gap. We guess it is because of the data: the public OpenWebText may have similar distribution to the closed-source data used to train GPT-2, but it may be short in number or quality. Another evidence is that none of the related work we cited successfully matches the original GPT-2 perplexities. For the SquaD results, we guess that it's a similar story, and code will be available to ensure reproductivity.
> - Training dynamics: please see the common response.
> - (Li et al., 2023) is summarized in Appendix (I).
> - Most Tables, figures, and minors are fixed in the new version, with a new citation to (Bartoldson et al., 2023) in Related Work. For the positions of Tables and figures, we will handle them in the final version.
>
> We hope that this solves some of your concerns and we are more than happy to have further discussions.

---

> > ### Comment · Reviewer_vhLH · 2023-11-23
> >
> > Thank you for these new, supporting results! Please consider adding a reference to Appendix J or K to the main text.
> >
> > I have improved my "soundness" rating and will recommend acceptance.

---

### Official Review · Reviewer_Bd3L · 2023-11-01

**Soundness:** 3 good
**Presentation:** 3 good
**Contribution:** 3 good
**Rating:** 6
**Confidence:** 3

**Summary:**

This paper proposes an interesting mechanism (masked structural growth) to accelerate the pre-training of foundation models. Technically, two main research problems are studied: i) how to determine the optimal growth schedule; and ii) how to design efficient growth operators. Some experiments are conducted to show the advances of the proposed method with a 2.2 times speedup.

**Strengths:**

- This paper studies a very important research problem in foundation model training. Such a technique can effectively reduce the end-to-end training time of the foundation models.

- Some theoretical analysis was presented to show the guidance of the design in the proposed method.

- The organization and presentation of the evaluation section are clear, with the hypothesis explicitly stated in each subsection.

**Weaknesses:**

- I am a little bit confused by the introduced method; intuitively, it seems to me that the proposed MSG operator would extend the parameter space to a more rugged space when compared with the average-based methods. Thus I cannot understand why it works well intuitively.

- The experimental results can be further improved:
  - The scale of the benchmarked model is small. Would the proposed method be deployed for a practice scale model, e.g., 7B to 13B or 13B to 30B?
  - The reported results can not demonstrate the real effectiveness of the proposed method in terms of ability to generalize, in fact, it seems that the results are mainly based on training loss while the most important metric should be a set of metric to measure the generalization ability of the model.
  - There should be some plot to show the training loss of the proposed method w.r.t different training iterations, this is essential to understand the performance.

**Questions:**

Please address my concerns listed in the Weaknesses section.

---

> ### Author Response · Authors · 2023-11-20
>
> Thanks for your insightful comments and suggestions!
> - An explanation of the effectiveness of MSG: (1) it introduces masks that make the process smoother for the model to adapt to the more “rugged” space caused by initialization; (2) average-based methods can underperform a naïve random initialization due to problems such as the Symmetry issue (Section 2.3.3). A more general speaking is: minimizing the difference to the original weights does not guarantee improving the training dynamics, since the latter is a research problem itself (also see Section 2.3.3). Another intuition (may be not so good) is that sometimes distillation works better than data-driven pruning for model compression.
>
> - Larger models: please see the common response.
> - Metrics: for the Bert models, we used an average of the GLUE scores. It already includes multiple metrics including Matthews correlation for CoLA, Pearson/Spearman correlation for STS-B and accuracy/f1 for MRPC, QQP, and SQuAD. Since these are a diversity of downstream NLP tasks, they would reflect some generalization ability of the models. The detailed results for each subtask are listed in Table 6 and 7. These may not be loss-based metrics since we observe that they are not strictly aligned with training loss in late phases (see footnote 4 and Figure 5). For GPT models, we follow common practices to report perplexity since they are aimed for text generation.
> - Training loss plot: please see the common response.
>
> We hope that this solves some of your concerns and we are more than happy to have further discussions.

---

> > ### Comment · Reviewer_Bd3L · 2023-12-04
> > **Thank you for your feedback**
> >
> > Thank you for your feedback! I adjusted my score.

---

### Author Response · Authors · 2023-11-20
**Common Response to Reviewers**

We would like to thank all the reviewers for their careful reading and constructive opinions.
- Common Concerns

(1) Scaling to Larger Models

Reviewers Bd3L and  aWRp raised questions about the method’s effectiveness on larger model sizes. This is indeed a significant topic. Actually, in the 4th paragraph in Introduction, we have discussed this problem by citing a “sanity check” report (Li et al., 2023) that builds a 101B model with growth operators oriented from MSG. We believe that this could provide you with some confidence that MSG is still effective when deployed on larger models. However, we agree that for large models, a systematic study on growth schedules and comparison to a full-size baseline is very expensive, thus we leave it to future works or community efforts. Finally, as required by another reviewer vhLH, we have summarized the technical report in Appendix (I).

(2) Loss Curve Visualization

Reviewers Bd3L and vhLH asked for a training loss curve plot. This is a good idea for an intuitive understanding of the growth strategy. We have added the plots in Appendix (K) in the last page. Please also check the new Figure 4. However, note that lower loss in late training does not necessarily indicate superior downstream performances, and thus the loss curves are just a qualitative indicator of the models’ capabilities.


- We have made the following modifications to the paper, please check the new version:

(1) Added Appendix (C) for some details of Bert2BERT.

(2) Added Appendix (I) to summarize the LLM technical report mentioned above.

(3) Added Appendix (J) for new GPT results after more sufficient training.

(4) Added Appendix (K) to visualize the training dynamics.

(5) Fixed typos, errors in equations, and the look of some Figure/Tables according to the reviewers' suggestions.


Some other issues are discussed in our comments to each reviewer. We are happy to receive these insightful suggestions and open to further discussion.

---

### Meta-Review · Area_Chair_3DUh · 2023-12-11

**Metareview:**

This paper studies approach for progressively growing Transformer structure starting from a smaller one. In particular, the paper proposes a new progressive training approach, Masked Structural Growth (MSG) with growth schedules (in multiple dimensions) and strictly function-preserving growth operators. The paper also provides empirical evidence suggesting that the approach significantly speeds up pretraining.

**Justification For Why Not Higher Score:**

There were concerns about the empirical evaluation scope and comparisons with the earlier works. At this point, several growth operators have been studied extensively and comprehensive comparison will be beneficial.

**Justification For Why Not Lower Score:**

The paper provides strong empirical results even though the results are on slightly smaller scale settings.

---

### Decision · Program_Chairs · 2024-01-16

Accept (poster)